# Regulator of Lipid Metabolism NHR-49 Mediates Pathogen Avoidance through Precise Control of Neuronal Activity

**DOI:** 10.3390/cells13110978

**Published:** 2024-06-04

**Authors:** Saebom Kwon, Kyu-Sang Park, Kyoung-hye Yoon

**Affiliations:** 1Department of Physiology, Yonsei University Wonju College of Medicine, Wonju 26426, Republic of Korea; saebom0504@naver.com; 2Mitohormesis Research Center, Yonsei University Wonju College of Medicine, Wonju 26426, Republic of Korea; 3Department of Global Medical Science, Yonsei University Wonju College of Medicine, Wonju 26426, Republic of Korea

**Keywords:** *Caenorhabditis elegans*, lipid metabolism, neuron, immune behavior, PPAR, *Pseudomonas aeruginosa*

## Abstract

Precise control of neuronal activity is crucial for the proper functioning of neurons. How lipid homeostasis contributes to neuronal activity and how much of it is regulated by cells autonomously is unclear. In this study, we discovered that absence of the lipid regulator *nhr-49*, a functional ortholog of the peroxisome proliferator-activated receptor (PPAR) in *Caenorhabditis elegans*, resulted in defective pathogen avoidance behavior against *Pseudomonas aeruginosa* (PA14). Functional NHR-49 was required in the neurons, and more specifically, in a set of oxygen-sensing body cavity neurons, URX, AQR, and PQR. We found that lowering the neuronal activity of the body cavity neurons improved avoidance in *nhr-49* mutants. Calcium imaging in URX neurons showed that *nhr-49* mutants displayed longer-lasting calcium transients in response to an O_2_ upshift, suggesting that excess neuronal activity leads to avoidance defects. Cell-specific rescue of NHR-49 in the body cavity neurons was sufficient to improve pathogen avoidance, as well as URX neuron calcium kinetics. Supplementation with oleic acid also improved avoidance behavior and URX calcium kinetics, suggesting that the defective calcium response in the neuron is due to lipid dysfunction. These findings highlight the role of cell-autonomous lipid regulation in neuronal physiology and immune behavior.

## 1. Introduction

Precise control of neuronal activity is essential for the proper functioning of the nervous system. Failure to maintain proper neuronal activity affects all aspects of their function, from development, behavior, learning, and memory, contributing to various neurological and neurodegenerative disorders [1,2,3,4,5].

Lipids play an important role in neuronal function as energy substrates, components of cellular structures, and bioactive molecules [6]. This importance has been underscored by the recognition that aberrant lipid metabolism in the nervous system is associated with neurological disorders [7,8,9,10,11]. However, how this regulation occurs within neurons remains underexplored. Contrary to most tissues, neurons lack lipid droplets and have limited capacity for fatty acid catabolism [12]. Energy production from fatty acid oxidation is instead carried out by astrocytes, generating lactate, which is then supplied to the neurons through the lactate shuttle [12]. Because of this, studies investigating lipid regulation in the brain either focus on lipid metabolism in the glia or do not differentiate where the regulation is occurring. The interdependence of neurons and glia in lipid-handling and metabolism poses a challenge in deciphering the distinct roles and contributions of how lipids are utilized and processed within each cell type. Therefore, in order to understand lipid regulation in neurons, studies disentangling the cell-autonomous and nonautonomous effects of lipid regulation are needed.

The nematode *Caenorhabditis elegans* has 284 nuclear hormone receptors (NHRs) [13]. Among these, NHR-49 functions as a regulator of lipid metabolism [14]. Initially identified as a key regulator of the fasting response, it controls the expression of many genes involved in fatty acid β-oxidation and desaturation, with functional similarity to PPARα [15,16]. The absence of NHR-49 results in pleiotropic defects, such as shortened lifespan and vulnerability to oxidative stress, hypoxia, and pathogens, demonstrating the importance of lipid metabolism in many diverse cellular and physiological processes [17,18,19,20,21]. However, despite NHR-49 being expressed in most cells, including neurons [16,22], how its function and thus lipid regulation in each tissue contribute to these overall defects has not been thoroughly examined.

Here, we describe a cell-autonomous role for NHR-49 in neurons in maintaining normal calcium activity in *C. elegans*. While investigating the role of NHR-49 in immunity, we found that the previously reported susceptibility of *nhr-49* mutants to the pathogen *Pseudomonas aeruginosa* PA14 was largely due to their defect in pathogenic lawn avoidance. Only restoring NHR-49 in the neurons, but not in the intestines or muscles, improved avoidance. Specifically, NHR-49 in three oxygen-sensing body cavity neurons, URX, AQR, and PQR, was sufficient to achieve avoidance behavior similar to the pan-neuronal rescue. Mutant analyses revealed that lowering the neuronal activity or ablating the body cavity neurons improved the avoidance response in *nhr-49* mutants. Calcium imaging of URX neurons revealed that *nhr-49* mutants displayed altered calcium kinetics in response to O_2_ stimulation, characterized by slower decay and a longer duration of calcium transients. The expression of NHR-49 in the body cavity neuron resulted in a decreased calcium transient duration. Lastly, supplementing with oleic acid also improved behavior and calcium activity. Taken together, this study shows that cell-autonomous regulation of lipid metabolism by NHR-49 is required to maintain normal calcium dynamics in *C. elegans* neurons.

## 2. Materials and Methods

### 2.1. C. elegans Growth and Maintenance

All strains used for the assay are listed in Appendix A. Strains were grown in nematode growth media (NGM) plates as previously described [23]. Strains were maintained at 20 °C, except for when worms were cultured in PA14 (25 °C).

### 2.2. PA14 Survival and Lawn Avoidance Assay

For both the survival and lawn avoidance assays, we followed previously established procedures [24]. A total of 7μL of overnight culture of PA14 in King’s B broth was seeded into slow-killing plates (NGM with 0.35% peptone). Plates were incubated at 37 °C for 24 h, then at room temperature for another 24 h. Thirty worms synchronized during the L4 stage were placed on PA14 lawns and maintained at 25 °C. Live worms were scored at 6–12 h intervals until all of them were dead. Three trials were conducted for each strain. Because worms were scored at slightly different time intervals between trials, representative plots are shown in the main text, and the rest are included in Appendix A (raw data). Lawn avoidance assays were prepared in the same way, and worms inside and outside the bacterial lawn were counted at indicated intervals. Later, we were able to record time-lapse videos of the plates overnight using an imaging system [25], which allowed us to score every hour, but we counted the worms at the same intervals to keep it consistent with previous trials. For lawn avoidance assays at 10% oxygen, plates were placed inside a hypoxia incubator chamber (Stemcell Technologies, Vancouver, BC, Canada) filled with 10% oxygen from the same pressurized gas tank used for calcium imaging. All the data shown in the same graph represent trials carried out side-by-side on the same day, using the same batch of PA14 plates and the same batch of age-synchronized worms.

### 2.3. Oleic Acid Supplementation

A total of 100 mM stock oleic acid (Sodium oleate, O7501, Sigma-Aldrich, St. Louis, MO, USA) was prepared in 3D.W. When preparing NGM plates, OA was added to a final concentration of 300 μM, along with 0.1% NP-40 to ensure even mixing of the lipids. Synchronized L1-stage worms were transferred to seeded plates and grown until the L4 (for the lawn avoidance assay) or young adult (for calcium imaging) larval stage.

### 2.4. Real-Time Quantitative PCR

Synchronized young adult worms were incubated in PA14 plates for 10 h. Worms were collected and washed with M9 buffer and frozen in TRIzol reagent (ZYMO Research, Irvine, CA, USA). RNA was extracted using a Direct-zol RNA miniprep kit (ZYMO Research, CA, USA). cDNA was constructed using the Maxime™ RT PreMix (Oligo (dT)15 Primer) (iNtRON Biotechnology, Seongnam, Republic of Korea). Quantitative PCR was conducted using the SYBR Green PCR Master Mix (Applied Biosystem, Waltham, MA, USA) on the QuantstudioTM6 Flex Real-Time PCR System (Applied Biosystem, Waltham, MA, USA). The primers used are listed in Appendix A.

### 2.5. Plasmid Constructs and the Generation of Transgenic Lines

Transgenic strains generated for this study are listed in Appendix A. Most of the strains generated for this study have been previously reported [26]. All primers used in this study that were not previously reported are listed in Appendix A. All plasmids were constructed using a previously generated plasmid that contained *nhr-49c*, followed by *SL2* and *gfp* or *mCherry* [26]. The 700 bp *gcy-32* promoter [27] was cloned and inserted into the plasmid using NotI and SmaI enzyme sites. The *nhr-49* gain-of-function alleles *et7*, *et8*, and *et13* were generated by site-directed mutagenesis of the *gcy-32p::nhr-49::sl2::gfp* plasmid using the Q5^®^ Site-Directed Mutagenesis Kit (NEB). GCaMP6s was cloned from the genomic DNA prep of the AML32 transgenic worm strain containing a pan-neuronal nuclear *GCaMP6s* construct. Primers amplified *GCaMP6s* without the nuclear localization signal sequence. The tetanus toxin light chain (tetx) sequence was cloned from the genomic DNA prep of the PS7200 transgenic worm strain. Plasmids were injected into *nhr-49(nr2041)* mutant worms at 30 ng/μL, along with the 40 ng/μL co-injection marker *unc-122p::mCherry*.

### 2.6. Calcium Imaging and Analysis

For calcium imaging, worms were immobilized in a poly(dimethylsioloxane) (PDMS) microfluidic chamber, as described previously [28]. Using the microfluidic chamber design generously provided by M. Zimmer, the SU-8 micro mold and PDMS chip were constructed by MicroFit (Hanam, Republic of Korea). The gas inlet of the microfluidic chamber was connected to two tanks of a pressurized gas mixture containing 21% O_2_ or 10% O_2_ via a three-way valve. The switch on the three-way valve allowed switching from one air source to another. The gas flow rate was set to 5 psi at the outlet of the gas regulator, and the flow rate entering the microfluidic chamber was adjusted to 600 cc/min using a flow meter (Dwyer, Michigan City, IN, USA). On Day 1, adult worms were transferred to unseeded plates to clean bacteria off the body and then transported to the chamber in S-basal buffer via polyethylene tubing (Intramedic, Sollentuna, Sweden). Worms were pre-exposed to 10% O_2_ for 5 min before imaging. GCaMP6s fluorescence was visualized at 40× magnification using an IX-73 inverted fluorescent microscope (Olympus, Tokyo, Japan) attached to an Iris 9 sCMOS camera (Photometrics, Tucson, AZ, USA) and a pE-800 LED illuminator (CoolLED, Andover, UK). Fluorescence was recorded using MetaFluor 3.1 at 2 frames/s, with the exposure time set to 80 ms. The resulting image sequence was processed using Fiji software (Version 2.14.0, NIH). The ROI was aligned using the ‘align slices in stack’ function of the Template Matching plugin, following which the pixel intensity of URX cell body was measured. GCaMP fluorescence in response to O_2_ tended to vary, and trials with a lower than 300% maximum ΔF/F_0_ were discarded and not used in the analysis.

## 3. Results

### 3.1. Neuronal NHR-49 Promotes Pathogen Avoidance to PA14

NHR-49 regulates many of the genes involved in lipid metabolism and is widely expressed in most tissues [14,16]. Although its absence results in pleiotropic defects, how NHR-49 functions in various tissues and, thus, lipid regulation in various tissues, mediates these defects is not understood. To explore the role of the tissue-specific function of NHR-49, we used transgenic strains that express NHR-49 under different tissue-specific promoters [26,29]. We wanted to see whether the expression of NHR-49 in any single tissue was sufficient to restore the known *nhr-49* phenotypes. Among the various known *nhr-49* phenotypes is their vulnerability to pathogens [19]. Therefore, we exposed each of the transgenic strains to *Pseudmonas aeruginosa* (PA14) to see if *nhr-49* was required in any specific tissue for its role in immunity. As previously reported, we found that *nhr-49* mutants show decreased survival in PA14 [20] (Figure 1A). Exposing transgenic rescue strains to PA14 revealed that, whereas intestinal NHR-49 improved survival, neuronal rescue did not (Figure 1B,C). Interestingly, we noticed a distinct difference in behavioral responses to the pathogens between wild-type N2 worms and *nhr-49* mutants. Wild-type worms display a bacterial lawn avoidance behavior to the harmful food source, moving outside the lawn within a few hours. However, *nhr-49* mutants remained inside the lawn and did not avoid the pathogenic bacteria (Figure 1D). When observed over 20 h, we found that mutants took significantly longer to leave the lawn than the wild type (Figure 1E).

To examine whether lawn avoidance behavior contributes to worm survival in PA14, we tested survival in a big lawn assay, where the PA14 lawn covers the entire agar surface, preventing worms from avoiding the pathogenic bacteria [30]. When unable to avoid, the difference in survival between N2 and *nhr-49* mutants became much smaller. This shows that the enhanced PA14 susceptibility seen in *nhr-49* mutants is not only due to their defect in immunity, but that much of it is also due to their defect in pathogen avoidance behavior (Figure 1F).

Pathogen avoidance behavior in *C. elegans* requires crosstalk between the nervous system and the intestines, the primary site of infection [31,32]. To find out if NHR-49 in a specific tissue is involved in mediating avoidance, we again tested avoidance with intestinal and neuronal, as well as muscle, NHR-49 rescue strains. Muscle-specific expression of *nhr-49* was shown previously to be ineffective in improving survival in PA14 [20]. Avoidance in intestinal and muscle NHR-49 rescue strains remained defective, but improved avoidance could be observed in the neuronal rescue strain (Figure 1G–I). Notably, none of the tissue-specific rescue strains of *nhr-49* were able to fully restore avoidance to wild-type levels, showing that NHR-49, in other tissues, may act together with neurons to achieve wild-type-level avoidance.

To narrow down which neurons are involved, we made transgenic strains expressing NHR-49 in specific subsets of neuron types. These include cholinergic, glutamatergic, serotonergic, and dopaminergic neurons. Among these, we found that NHR-49 in the cholinergic and glutamatergic neurons was sufficient to restore avoidance to that observed in the pan-neuronal rescue (Figure 2C,D). NHR-49 in serotonergic and dopaminergic neurons did not improve avoidance, even though several serotonergic neurons are also cholinergic (Figure 2A,B). In addition, contrary to the pan-neuronal rescue, which showed improved avoidance but no change in survival, the cholinergic and glutamatergic rescue strains both showed increased survival (Figure 2E,F).

### 3.2. NHR-49 Acts Downstream of the TGFβ/DAF-7 Pathway

Several signaling pathways and neuronal circuits that contribute to PA14 lawn avoidance have been identified in previous studies (Figure 3A). It is known that the detection of PA14 metabolites causes an upregulation of the TGFβ ligand DAF-7 in the ASJ sensory neuron, activating the downstream TGFβ pathway [24]. To test whether the avoidance defect was due to defective TGFβ signaling, we exposed worms to PA14 and tested the increase in *daf-7* by real-time PCR. In the wild-type strain, the 10 hr exposure time we used was too short to reliably detect *daf-7* increase (Figure 3B). However, *daf-7* consistently increased in the *nhr-49* mutant, showing that upregulation of DAF-7 in response to PA14 occurs normally, if not more strongly, in the *nhr-49* mutant (Figure 3B). DAF-3 is a co-Smad protein located downstream of DAF-7 and is a negative regulator of the pathway. Mutants of *daf-3* can suppress any defects that occur upstream [24]. Whereas *daf-3* showed normal avoidance as expected, the *daf-3;nhr-49* double mutant still showed delayed lawn avoidance behavior, indicating that *nhr-49* is acting downstream or parallel to the TGFβ pathway (Figure 3C).

### 3.3. NHR-49 Acts Downstream of NPR-1 Ligand Upregulation

Another pathway that has been studied regarding PA14 avoidance is the activation of immunity through the neuropeptide GPCR NPR-1 [30,33,34]. The neuropeptides FLP-18 and FLP-21 are NPR-1 ligands whose expression is upregulated in response to PA14 infection [33]. Assessing ligand expression levels using qPCR showed that *flp-18* and *flp-21* are upregulated in both wild-type and *nhr-49* mutants, with *nhr-49* showing an even greater increase of the two genes compared to the wild type (Figure 3D). Therefore, upregulation of the NPR-1 ligands occur normally, if not better, in *nhr-49* mutants, suggesting *nhr-49* is acting downstream or parallel to the NPR-1 ligands.

### 3.4. NHR-49 in URX, AQR, and PQR Neurons Are Sufficient for Pathogen Avoidance

NPR-1 is expressed in several neurons, but NPR-1 in the oxygen-sensing body cavity neurons AQR, PQR, and URX were previously shown to be involved in innate immunity and pathogen avoidance [30,34]. Activation of NPR-1 causes cell-nonautonomous upregulation of immunity genes, including the activation of the p38 MAPK pathway [34]. Since they represent the most downstream neurons known to act in immunity and pathogen avoidance, we expressed NHR-49 in these neurons. Notably, AQR and PQR are glutamatergic neurons while URX is cholinergic, the two neuron types where the presence of NHR-49 was shown to improve avoidance (Figure 2C,D). Strikingly, we found that expressing NHR-49 in the three neurons was sufficient to improve avoidance to a level similar to that displayed by the pan-neuronal strain (Figure 3E). The strain also showed improved survival in the pathogenic bacteria (Figure 3F).

NHR-49 is a nuclear hormone receptor, which is a transcription factor activated by lipid ligands. Three gain-of-function alleles for *nhr-49* have previously been identified [35]. These gain-of-function alleles are each known to increase NHR-49 activity to varying degrees and affect binding to interacting partners such as NHR-66, resulting in similar but distinct changes in the expression of downstream target genes [35]. We expressed each of the gain-of-function alleles in the body cavity neurons and found that the *et7* and *et8* alleles further improved avoidance compared to the wild-type allele, while *et13* made no difference (Figure 3G–I). This indicates that *nhr-49* expression, NHR-49 activity, and/or the NHR-49 target genes affect how body cavity neurons respond to pathogens.

### 3.5. Body Cavity Neuron Activity Negatively Contributes to Pathogenic Lawn Avoidance

What role could NHR-49 have in the body cavity sensory neurons to promote pathogen avoidance behavior? Past studies have shown that oxygen sensation is closely associated with immunity and pathogenic lawn avoidance in *C. elegans*. In addition to being the receptor for neuropeptide signaling immunity [30,34], NPR-1 affects organismal and cellular responses to oxygen gas [36]. The defective pathogenic avoidance phenotype of *npr-1* loss-of-function mutants is attributed to hyperoxia avoidance, where worms prefer to stay inside the lawn where oxygen remains low due to bacterial respiration [37]. Thus, when the ambient oxygen concentration is lowered, *npr-1* mutant worms are able to leave the pathogenic lawn. Likewise, a mutation in the oxygen-sensing guanylyl cyclase also allows the *npr-1* mutants to leave the lawn [32].

We wondered whether the relationship between oxygen and pathogen avoidance could also be observed in *nhr-49* mutants. To find out, we first tested PA14 lawn avoidance in 10% ambient oxygen. We found that lower oxygen levels did in fact improve the avoidance of *nhr-49* mutants to levels similar to that of the wild type (Figure 4A). On the other hand, 10% oxygen did not significantly alter avoidance in N2 worms. We also generated a double mutant of *nhr-49* and the oxygen-sensing guanylyl cyclase *gcy-36* and found that the double mutant displayed improved lawn avoidance (Figure 4B).

Interestingly, previous studies have shown that the pathogen survival of the *npr-1* mutant improves when the oxygen-sensing AQR, PQR, and URX are ablated [34]. We wondered whether the ablation of the same neurons would improve the pathogen avoidance of *nhr-49* mutants as well. We crossed *nhr-49* mutant worms with a transgenic strain whose body cavity neurons were genetically ablated by expressing the cell-death activator *egl-1* [34]. We found that genetic ablation of AQR, PQR, and URX neurons substantially improved avoidance in both the wild type and *nhr-49* (Figure 4C).

Next, we wondered whether abolishing synaptic transmission would have the same effect as neuronal ablation. We generated a transgenic strain that expresses the tetanus toxin light chain (tetx), which silences synaptic transmission by cleaving synaptobrevin under the body cavity neuron-specific promoter [38]. We found that preventing synaptic transmission in the body cavity neuron improved avoidance in the wild type, but it failed to improve avoidance in the *nhr-49* mutant (Figure 4D). Thus, synaptic transmission from body cavity neurons is likely not involved in the defective pathogen avoidance of *nhr-49* mutants.

### 3.6. NHR-49 Affects URX Calcium Kinetics

Although these results may indicate that *nhr-49* mutants have altered oxygen preferences and hyperoxia avoidance, there was no indication of this in their behavior. *npr-1* mutant worms display very distinct behaviors due to their oxygen preference, such as bordering behavior, in which worms prefer to stay at the thickest part of the lawn, or social feeding behavior, in which worms aggregate at these borders (Appendix A) [36]. However, even by just casual observation, *nhr-49* mutants displayed none of these behaviors (Appendix A).

We also noted that although restoring *nhr-49* expression in the body cavity neurons improved avoidance, the absence of these neurons altogether was even more effective (Figure 2E and Figure 4C). To make sense of this, we wondered whether restoring *nhr-49* expression in these neurons caused them to be less active. When seen in this way, pathogenic lawn avoidance can be explained by the neuronal activity of body cavity neurons; the activity of these neurons negatively impacts pathogen avoidance. Manipulations that lower activity, such as low ambient oxygen, or the absence of the oxygen-sensing guanylyl cyclase improve avoidance. Ablating the neurons abolishes activity as well as their synaptic connections in the larger neuronal circuit and thereby improves avoidance. Conversely, avoidance-defective *npr-1* mutants are known to display longer and sustained tonic activity in response to O_2_ upshifts [39]. In this vein, we hypothesized that the absence of *nhr-49* could prevent lawn avoidance by increasing the body cavity neuron activity, while restoring *nhr-49* expression lowers it closer to the wild type.

To consider this possibility, we expressed GCaMP6s in the URX neuron to image its calcium activity in response to an O_2_ upshift (Figure 5A). As reported previously, an increase in O_2_ concentration caused a fast rise and fall in calcium in wild-type URX [28]. Applying the same stimulation to *nhr-49* mutants resulted in calcium transients with similar amplitudes (Figure 5B,E). However, the calcium transients in *nhr-49* mutants seemed to show trends of slower decay compared to the quick exponential decline seen in the wild type (Figure 5A,B,G). To compare the kinetics of the calcium transients between the wild-type and *nhr-49* mutants, we characterized the properties of the calcium transients in more detail, such as the peak amplitude (F_max_), half time for rise and decay (t_1/2_ rise and t_1/2_ decay, respectively), area under the curve, and transient duration at 10%, 50%, and 80% repolarization (TD10, TD50, and TD80, respectively) (Figure 5D–I). From this, we found that the t_1/2_ decay time of *nhr-49* indeed showed a significant increase, as well as a slight increase in the t_1/2_ rise time (Figure 5F,G). These results show that calcium rises and falls more slowly in the mutant. Moreover, calcium transients in the mutant showed longer durations at 10, 50, and 80% repolarization (Figure 5I). Thus, URX calcium transients responding to O_2_ upshifts are overall longer in duration.

We reasoned that if this subtle change in calcium kinetics is functionally meaningful, we would see these parameters return closer to the wild type when avoidance is restored. Therefore, we imaged URX calcium activity in the body cavity neuron rescue strain, which previously showed improved avoidance (Figure 3E). We found that, indeed, the transgenic strain showed a trend towards decreased t_1/2_ decay and transient duration, though not statistically significant (Figure 5G,I). We noted that just as in behavior, it did not completely restore the kinetics back to the wild type. Thus, the absence of *nhr-49* in the body cavity neurons leads to slower calcium decay and a longer transient duration that corresponds to the pathogen avoidance behavior.

### 3.7. Restoring Lipid Homeostasis Is Sufficient to Improve Pathogen Avoidance

How is NHR-49 mediating neuronal activity? NHR-49 is known to regulate lipid metabolism, and *nhr-49* mutants show increased ratios of saturated to unsaturated fatty acids [16]. For some phenotypes, the NHR-49 defect can be ameliorated by supplementation with the 18-carbon monounsaturated fatty acid, oleic acid (OA) [40]. We wondered whether supplementation with OA would also improve pathogen avoidance in *nhr-49* mutants. Indeed, when *nhr-49* mutant worms were grown in 300 μM OA, they showed similarly improved lawn avoidance to the pan-neuronal or body cavity neuron NHR-49 rescue strains (Figure 5J). Moreover, we tested whether supplementing OA can further enhance avoidance of the body cavity neuron rescue strain, but found no further improvement. This suggested the possibility that OA supplementation and NHR-49 in body cavity neurons are acting through the same mechanism to improve avoidance (Appendix A). Next, to see if OA supplementation restored calcium kinetics of the URX neuron, we imaged the URX calcium activity of OA-supplemented worms. We found that the duration of calcium transients also showed a decreasing trend compared to the untreated mutant (Figure 5I,K). Thus, restoring lipid balance by OA supplementation improves behavior and calcium kinetics, indicating that lipid dysfunction in the body cavity neurons contributes to calcium mishandling and lawn avoidance defects.

## 4. Discussion

NHR-49 is a well-established lipid regulator in *C. elegans* with functional similarities to PPARα [14]. Mutants show altered lipid compositions, most prominently, an increased saturated-to-unsaturated fatty acid ratio [16,40]. The requirement of NHR-49 in various *C. elegans* longevity models, such as the germline-less *glp* mutants, vitellogenin overexpression, dietary restriction (DR), and increased lysosomal lipid signaling, point to its role in mobilizing and utilizing lipids [16,41,42,43,44]. Although its expression is observed in most cells, studies on its role in neurons have been lacking. However, a few recent studies are beginning to uncover NHR-49’s function in neurons: neuronal NHR-49 was shown to increase lifespan [20,26,29], and it was also shown to be required for neuropeptide release in response to lipid signals from the intestine [45].

Our work shows that NHR-49 in neurons cell-autonomously contribute to the precise regulation of calcium activity and neuronal function. Altered calcium activity, namely prolonged calcium responses, in the URX neuron correlated with defective pathogen avoidance. Both calcium activity and pathogen avoidance were ameliorated either by restoring *nhr-49* in the neuron or by supplementing OA to restore lipid homeostasis.

Although NHR-49 undoubtedly contributes to immunity against PA14, especially through its expression in the intestines, how it contributes to pathogen behavior was previously unrecognized. Our work shows that a large part of the pathogen susceptibility seen in the *nhr-49* mutant is due to its behavioral defect, and that survival is improved if worms are able to avoid the pathogen. The exception was the pan-neuronal rescue strain, which showed improved avoidance but no change in survival. Considering that different types of neurons are known to either promote or suppress immunity [46,47], and that survival is the combined result of immunity and behavior, this may indicate that NHR-49 in different neurons contribute to antagonistic effects in immunity. Interestingly, others have reported improved survival when *nhr-49* was restored pan-neuronally [20]. This could be due to the different promoters used for pan-neuronal expression: a previous study used the *unc-119* promoter, whereas we used the *rgef-1* promoter, two of the most commonly used pan-neuronal promoters in *C. elegans* studies. Different promoters are known to result in different levels of expression in various neuronal cell types, which may explain this discrepancy. Altogether, this highlights the importance of the behavioral aspect of immunity when evaluating pathogenic infections.

While the change in calcium signal resulting from the absence of *nhr-49* is subtle, this subtle difference was proportionate to the level of behavioral defect displayed by the mutant. Whereas *npr-1* mutants, which respond to O_2_ upshifts with a sustained calcium activity, are unable to completely avoid PA14 even after 20 h of PA14 exposure [25,33], *nhr-49* mutants do eventually leave the lawn. It can be argued that this corresponds to the mildly prolonged calcium response seen in *nhr-49* mutants. In addition, calcium kinetics in the genetic rescue or OA-supplemented worms do not completely recover to wild-type levels, just as behavior does not quite fully recover. Furthermore, it is possible that the subtle change we see in the whole-cell calcium activity reflects a change in calcium activity in a certain microdomain that was not discernable in our analysis.

Slower decay of calcium transients can be an indication of defective calcium clearance [48]. After a calcium influx, cytoplasmic calcium is removed by ATP-dependent mechanisms. This includes plasma membrane Ca^2+^ ATPase (PMCA), which extrudes Ca^2+^ to the extracellular space, or sarco/endoplasmic reticulum Ca^2+^ ATPase (SERCA), which pumps Ca^2+^ into the endoplasmic reticulum (ER) lumen. Whether the prolonged Ca^2+^ transient in *nhr-49* mutants is due to defects in any specific mechanisms or to a general decline in Ca^2+^ regulation remains to be seen.

We have previously reported evidence of altered neuronal activity in another neuron in the *nhr-49* mutant—the HSN motor neuron, which controls egg-laying [26]. Mutants lay earlier-stage embryos, which is usually an indication of increased HSN activity and vulva muscle contractions. Restoring *nhr-49* in three serotonergic neurons, which included the HSN motor neuron, caused worms to lay later-stage eggs comparable to the wild-type. Taken together, these pieces of evidence suggest the possibility that *nhr-49* is involved in cell-autonomous regulation of neuronal activity in other neurons as well.

Lipids are involved in many aspects of neuronal calcium signaling by either directly interacting with the channels or by forming different lipid microdomains that affect channel localization or function [49]. Their importance is illustrated by the fact that many neurodegenerative diseases show altered lipid metabolism, along with altered calcium signaling [48]. For example, the activity of plasma membrane Ca^2+^ ATPase (PMCA), which extrudes cytosolic Ca^2+^ to the extracellular space, is regulated by acidic phospholipids [50]. In addition, various phospholipids and sphingolipids, as well as cholesterol, regulate the gating of transient receptor potential (TRP) channels and voltage-gated calcium channels [49]. Lipids also function as secondary messengers, such as in the cases of inositol-3-phosphate (IP3) and diacylglycerol (DAG). Accordingly, studies in *C. elegans* have also shown that the polyunsaturated fatty acids (PUFAs) omega-3 and -6 fatty acids modulate neuronal activity [51], interact with ion channels, such as TRPV [52,53,54], and are required for efficient neurotransmission and signal transduction [55]. The Δ6 desaturase *fat-3* mutants, which lack 20-carbon omega-3 and omega-6 fatty acids, have various behavioral defects such as defective motor coordination, fecundity, and sensory functions such as chemotaxis, odor learning, response to noxious stimuli, and mechanosensation [52,53,54]. These defects could be rescued by supplementing omega-3 or -6 fatty acids. In one of these studies, calcium responses in *fat-3* ASH neurons were recorded, which showed diminished calcium responses to glycerol. Supplementing the omega-3 fatty acid eicosapentaenoic acid (EPA) was sufficient to rescue the calcium response [52]. Since *fat-3* is expressed in many tissues, including neurons, it would be interesting to see whether *fat-3* is required in the neuron itself or whether EPA can be transported from other tissues expressing *fat-3*.

The current study raises the question of which downstream targets of NHR-49 contribute to calcium activity in neurons. Many of the NHR-49 downstream target genes have roles in fatty acid β-oxidation and desaturation, but since β-oxidation is known to occur negligibly in neurons, whether neuronal NHR-49 functions through the same targets is unclear. In addition, GFP reporter strains indicate that many of the target genes, such as acyl-coA synthetase *acs-2*, and Δ9 desaturase *fat-6* and *fat-7*, are most prominently expressed in the intestines, the main metabolic organ of *C. elegans* [56]. However, this does not preclude the possibility of genes being expressed in neurons in lower levels. Also, since the binding of NHRs to their target sequences is dependent on the availability of different ligands and binding partners [57], NHR-49 may have different target genes in neurons than in other tissues. Interestingly, among the three gain-of-function alleles, the *et7* and *et8* alleles, but not *et13*, further improved lawn avoidance. Comparing the downstream targets between the different gain-of-function alleles may provide a clue as to which of the downstream genes are involved in regulating neuronal activity.

It is noteworthy that although an exogenous supply of OA alleviated behavior and calcium activity, the expression of NHR-49 in the intestines, which is viewed as the main metabolic tissue in *C. elegans*, did not improve avoidance. This suggests that OA synthesis in the intestines is not sufficient to supply OA to the neurons. Perhaps similarly, both neuron-specific knockdowns of the sterol regulatory element binding protein (SREBP) gene, an important regulator of fatty acid production, resulted in defective dendrite arborization in the mechanonociceptive class IV da neuron of the *Drosophila* larvae [58]. This defect was not observed when the same gene was knocked down in neighboring glia or epidermal cells, showing that the effect is cell-autonomous [58]. Animals with defective arborization were hypersensitive to mechanical stimuli, which again suggests altered calcium activity [58].

Because they regulate genes involved in similar processes, NHR-49 is often compared to PPARα. Mammalian PPARs—α, β, and γ—are expressed widely throughout the brain, each with their own distinct pattern of distributions. PPARα expression is found in both neuronal and glial populations, whereas PPARβ and PPARγ are found more specifically in neuronal populations [59]. While PPARα deletions in mice do not result in overt defects in locomotion, brain development, and function [60], additional studies identified deficits in learning and memory [61], as well as synaptic function and plasticity [62,63,64]. In addition, ligand-mediated activation of PPAR is reported to be protective against neurodegeneration [65,66,67]. Whether these effects are mediated by PPAR in the neurons or in the glia are not yet known. Identifying the downstream target genes responsible for these effects will shed light on the various ways lipids regulate neuronal function and may help identify potential therapeutic targets.

## 5. Conclusions

Our study shows that the expression of the lipid regulator NHR-49 acts cell-autonomously to ensure proper neuronal activity. The absence of *nhr-49* in the body cavity neurons results in neuronal activity that is longer in duration, significantly altering the behavior of the whole organism to the pathogenic bacteria PA14. Further studies to identify the downstream target genes will help elucidate the mechanisms of such defects. As aberrant lipid metabolism is often implicated in neurological and neurodegenerative disorders, further understanding of this process may help identify potential therapeutic targets.

## Figures and Tables

**Figure 1 cells-13-00978-f001:**
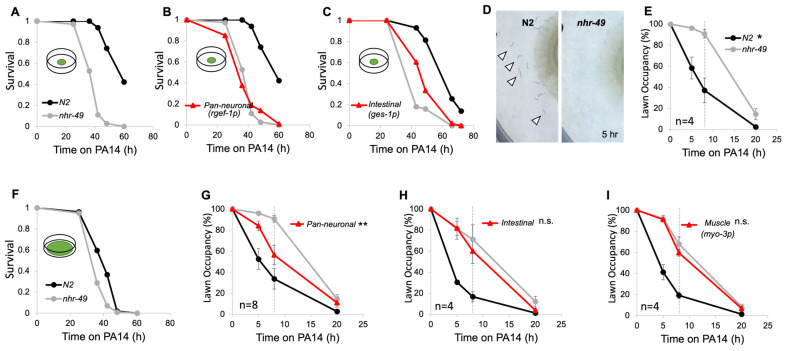
*nhr-49* mutants exhibit decreased survival and defective pathogenic lawn avoidance behavior against PA14. (**A**–**C**) Representative survival plots in small lawns. Three trials were conducted for all survival assays (See Appendix A for additional trials). (**D**) After 5 h of PA14 exposure, wild-type worms are seen outside the lawn (arrowhead), whereas *nhr-49* mutants remain inside the lawn. (**E**) Lawn avoidance of N2 and *nhr-49.* The dotted gray line at the 8 h mark indicates that statistical analyses were conducted using the data for 8 h post-exposure. (**F**) Representative survival plots of big lawn survival assay (See Appendix A for additional trials). (**G**–**I**) Lawn avoidance of tissue-specific transgenic strains. The dotted gray line at the 8 h mark indicates that statistical analyses were conducted using the data for 8 h post-exposure. For statistical analyses of avoidance assays, each strain was compared to *nhr-49* (one-way ANOVA with Dunnett’s multiple comparisons test). Asterisks indicate the *p*-value (* *p* < 0.05; ** *p* < 0.01).

**Figure 2 cells-13-00978-f002:**
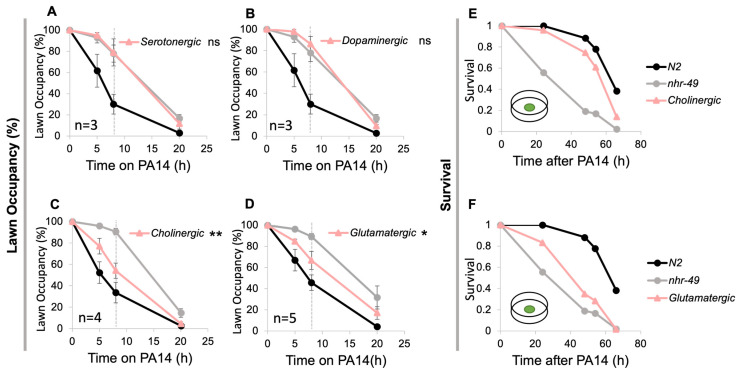
NHR-49 in cholinergic and glutamatergic neurons promote pathogenic lawn avoidance. Lawn avoidance of serotonergic (**A**), dopaminergic (**B**), cholinergic (**C**), and glutamatergic (**D**) neuron rescue strains. (**E**,**F**) Representative survival plots for cholinergic and glutamatergic rescue strains. Plate with small green lawn indicates survival was tested in a small lawn assay. Three trials were conducted for all survival assays (See Appendix A for additional trials). For lawn avoidance (**A**–**D**), the dotted gray line at the 8 h mark indicates that statistical analyses were conducted using the data for 8 h post-exposure. Each strain was compared to *nhr-49*. (one-way ANOVA with Dunnett’s multiple comparisons test). Asterisks indicate the *p*-value (* *p* < 0.05; ** *p* < 0.01).

**Figure 3 cells-13-00978-f003:**
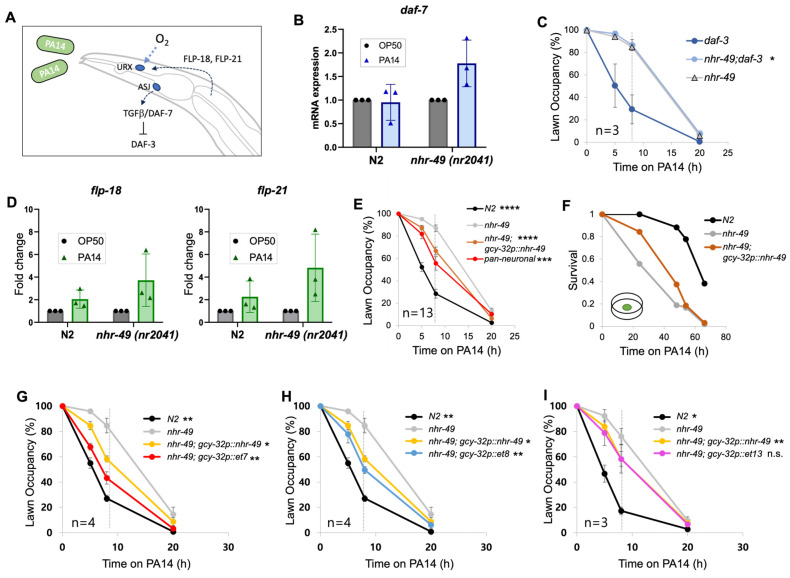
NHR-49 in the URX, AQR, and PQR body cavity neurons is sufficient to promote avoidance. (**A**) Signaling pathways involved in promoting pathogenic lawn avoidance. (**B**) Real-time PCR quantification of *daf-7* mRNA expression after 10 h of PA14 exposure. (**C**) The lawn avoidance defect of *nhr-49* is not suppressed by *daf-3* mutations. (**D**) Real-time PCR quantification of NPR-1 ligands *flp-18* and *flp-21* mRNA expression after 10 h of PA14 exposure. (**E**,**F**) Body cavity neuron-specific rescue improves lawn avoidance (**E**) and survival (**F**). Three trials were conducted for survival (see Appendix A for additional trials). (**G**–**I**) Lawn avoidance in strains with body cavity neuron-specific expression of *nhr-49* gain-of-function alleles. The dotted gray line at the 8 h mark indicates that statistical analyses were conducted using the data for 8 h post-exposure. Each strain was compared to *nhr-49*, except (**C**), which was compared with *daf-3* (one-way ANOVA with Dunnett’s multiple comparisons test). Asterisks indicate the *p*-value (* *p* < 0.05; ** *p* < 0.01; *** *p* < 0.001; **** *p* < 0.0001).

**Figure 4 cells-13-00978-f004:**
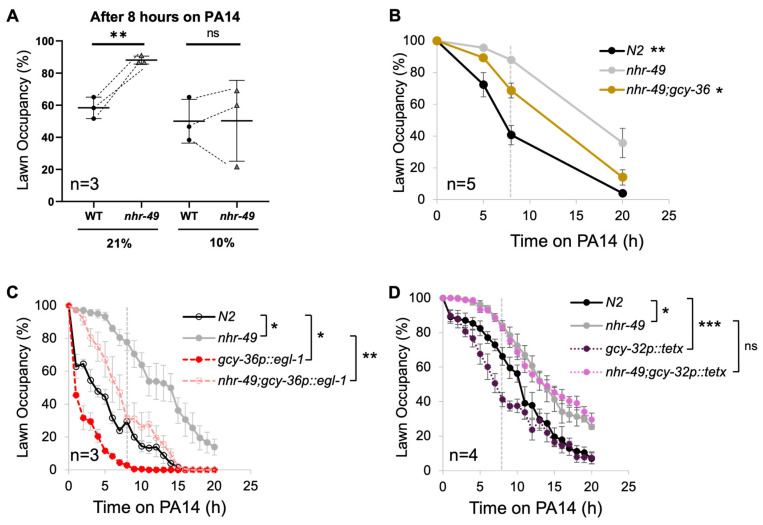
Decreased activity or ablation of body cavity neurons leads to improved avoidance in *nhr-49* mutants. (**A**) PA14 avoidance in 21% and 10% ambient oxygen after 8 h. (**B**) Pathogen avoidance of oxygen-sensing defective mutants. (**C**) Genetic ablation of body cavity neurons results in improved avoidance in both wild-type and *nhr-49* mutants. (**D**) Abolishing the synaptic signal does not improve avoidance. The dotted gray line at the 8 h mark indicates that statistical analyses were conducted using the data for 8 h post-exposure. Each strain was compared to *nhr-49* (one-way ANOVA with Dunnett’s multiple comparisons test for (**B**); Student’s *t*-test for (**C**,**D**)). Asterisks indicate the *p*-value (* *p* < 0.05; ** *p* < 0.01; *** *p* < 0.001).

**Figure 5 cells-13-00978-f005:**
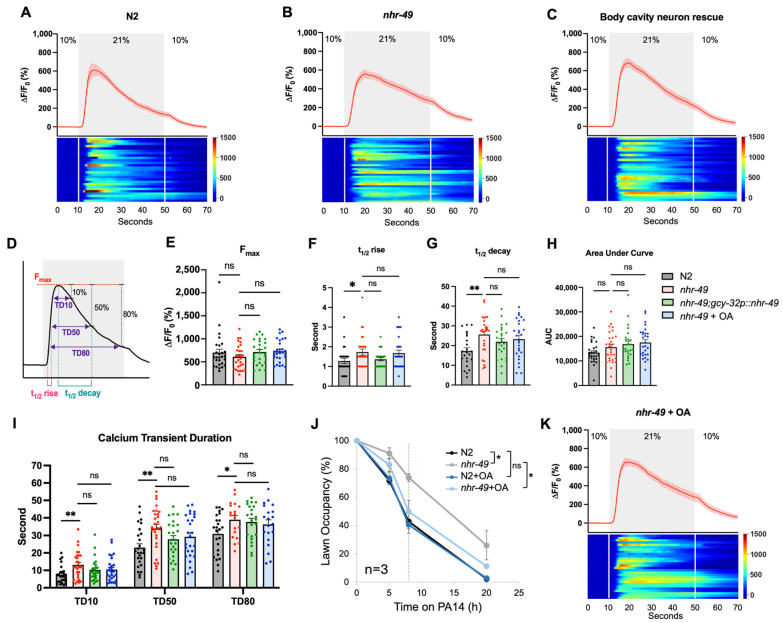
Calcium transients of *nhr-49* show prolonged signals. (**A**–**C**) Calcium imaging results of the N2, *nhr-49*, and body cavity rescue strains in response to an O_2_ upshift from 10% to 21%. (**D**–**I**) Detailed characterization of calcium transients. Peak amplitude (**E**), half time of rise and decay (**F**,**G**), area under the curve (**H**), and transient duration at 10, 50, and 80% repolarization (**I**) were assessed. (**J**) Avoidance behavior of N2 and *nhr-49* mutants supplemented with 300 μM OA. The dotted gray line at the 8 h mark indicates that statistical analyses were conducted using the data at this time point. (**K**) Calcium imaging results of *nhr-49* supplemented with OA. Asterisks indicate the *p*-value (* *p* < 0.05; ** *p* < 0.01), as determined by the one-way ANOVA with Dunnett’s multiple comparisons test for (**E**–**I**) and the Student’s *t*-test for (**J**).

## Data Availability

Raw data for Figure 1, Figure 2, Figure 3, Figure 4 and Figure 5 are provided in Appendix A.

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
