# Peer review of "Regulator of Lipid Metabolism NHR-49 Mediates Pathogen Avoidance through Precise Control of Neuronal Activity"

_cells, 2024, doi:10.3390/cells13110978_

Round 1

Reviewer 1 Report

Comments and Suggestions for Authors

Cell -2974688 review

Manuscript title: “Regulator of Lipid Metabolism NHR-49 Mediates Pathogen Avoidance through Precise Control of Neuronal Activity”, The importance of research on nuclear hormone receptors (NHRs) lies in their central role in various biological processes and in their potential as targets for therapeutic intervention. By modulating the activity of NHRs through ligands, researchers can influence gene expression and thereby affect a variety of physiological functions, including development, metabolism, stress response and aging. One notable aspect of NHRs is their ability to regulate lipid metabolism, which is critical for cellular homeostasis and overall organismal health. In the case of NHR-49 in C. elegans, its cell-autonomous regulation of lipid metabolism has been shown to be essential for the maintenance of normal calcium dynamics in neurons. This finding underscores the intricate interplay between lipid metabolism and neuronal function and highlights the diverse roles of NHRs in the physiology of the organism.

The objectives of the study clearly stated.

The study methods (applicability/models C. elegans) reported in sufficient to allow for their replicability (consist according to publication early Kwon S, Park KS, Yoon KH. Dissecting the Neuronal Contributions of the Lipid Regulator NHR-49 Function in Lifespan and Behavior in C. elegans. Life (Basel). 2023 Dec 15;13(12):2346. doi: 10.3390/life13122346. PMID: 38137948; PMCID: PMC10744624)

The statistical analyses used, controls, sampling mechanism, and statistical are reporting appropriate.

The discussion section, have the authors clearly emphasized the strengths of their study.

The manuscript structure (e.g., sections and its subheading), text flow are writing appropriate.

The work is scientifically sound and worthy of consideration for publication. 

Author Response

Thank you for your positive review.

Reviewer 2 Report

Comments and Suggestions for Authors

In the manuscript entitled “Regulator of lipid metabolism NHR-49 mediates pathogen avoidance through precise control of neuronal activity” by Kwon et al., the authors investigated the role of one of the most well-characterized nuclear hormone receptors in C. elegans, NHR-49, in mediating pathogen avoidance behavior. C. elegans has been shown to avoid the pathogen Pseudomonas aeruginosa (PA14) and this behavior is regulated by the nervous system. The authors discovered that this pathogen avoidance behavior depends on the presence and activity of NHR-49 in oxygen-sensing body cavity neurons URX, AQR and PQR. Oleic acid supplementation improved the pathogen avoidance behavior of the NHR-49 mutant worms suggesting lipid dysfunction as the main reason underlying impaired behaviors of these mutant worms. Overall, the manuscript is written well, and results are interpreted correctly. My specific comments are summarized here.

1.   In multiple occasions, pathogen avoidance behavior does not coincide with the ability of the worms to survive on PA14 lawn. For example, transgenic worms expressing NHR-49 in the intestine survive better than the mutant worm but doesn’t show any difference in behavior. This could be intriguing, but the possible reasons should be discussed in more detail.

2.      Y axis should be labelled individually in all sub-figures.

3. The order of the sub-figures should follow their appearance/mentioning in the text. This is violated multiple times. For example, Figure 1D was mentioned after Figure 1E/1F.

4.     What is the survival of daf-3;nhr-49 double mutant on PA14 lawn?

5.      Supplementary Figure 1. The figure legend is missing.

6.  Figure 4A. How many times this experiment was repeated? That   information is missing.

7.       Figure 4A. Font size of the axis labels should be uniform.

8.       Figure 3I is missing from the figure legend.

9.       Reference is missing in line 303.

10. There are some typographical and grammatical errors in the  manuscript  that need to be corrected.

Reviewer 3 Report

Comments and Suggestions for Authors

This manuscript by Kwon S. et al. investigates the cell-autonomous role of NHR-49, a master regulator of lipid homeostasis, in regulating neuronal activity and pathogen avoidance in Caenorhabditis elegans. The study demonstrates that neuron-specific expression of NHR-49 can rescue the impaired avoidance behavior of nhr-49 mutants against the pathogen Pseudomonas aeruginosa (PA14 strain). Notably, the restoration of NHR-49 function in specific oxygen-sensing body cavity neurons not only rescues pathogen avoidance behavior but also improves neuronal calcium kinetics. The supplementation with oleic acid underscores the link between lipid metabolism impairment and neuronal activity, suggesting potential therapeutic interventions.

Major Comments:

1.        The manuscript presents conflicting results regarding pan-neuronal and tissue-specific expression of NHR-49 in relation to survival and pathogen avoidance. While the results convincingly demonstrate that PA14 avoidance behavior necessitates neuronal expression of NHR-49, it is puzzling that pan-neuronal expression, driven by the rab-3 promoter, does not enhance survival against PA14 exposure as shown in Figure 1B. Notably, the rab-3 promoter has been reported to exhibit leaky expression in the intestine, which might influence these results (Zhang et al., 2022 Nat Commun, 13(1): 6339). Moreover, it is intriguing that specific expression of NHR-49 in cholinergic and glutamatergic neurons improves survival (Figures 2E and 2F), while pan-neuronal expression does not. This discrepancy raises the possibility that NHR-49 may have antagonistic roles in different neuronal types. Moreover, it is widely accpeted that although rab-3, rgef-1, snb-1, unc-119 are considered as pan-neuronal promoters they are not expressed in 302 neurons in the transgenci animals. This is happening due to mosaicisms or different expression levels of the promoters in diverese neuronal cell types. To resolve these inconsistencies and confirm the findings, it would be prudent for the authors to replicate the survival experiments using another pan-neuronal promoter, such as rgef-1 or snb-1, which may offer a more definitive expression profile.

2.        The absence of data on the effects of muscle-specific expression of NHR-49 on survival is a significant omission, given that information on avoidance behavior is provided. Including this data could offer a more comprehensive understanding of NHR-49’s role across different tissues.

3.        The auhtors attempt to correlate pathogen avoidance with survival, yet this correlation appears incomplete or inconsistent. For instance, intestinal rescue enhances survival without improving avoidance behavior. Clarification and additional experimental validation of these observations are essential to strengthen the study’s conclusions.

4.        The authors should concider the effects of oleic acid supplementation not only on nhr-49 mutants but also on strains expressing NHR-49 in specific neurons could provide insights into potential non-cell-autonomous effects of lipid regulation. In Figure 5J, the authors supplemented animals with oleic acid, known to mitigate some effects of NHR-49 depletion. Their results convincingly demonstrate that oleic acid supplementation substantially restores the impaired avoidance behavior of nhr-49 mutants. However the authors need to evaluate the impact of oleic acid supplementation on animals expressing NHR-49 in the three specific neurons (URX, AQR, and PQR), where it has been shown to moderately improve avoidance behavior (Figure 3F). An enhanced response to oleic acid in these transgenic lines could suggest contributions from other tissues, particularly the intestine, highlighting potential non-cell-autonomous effects of lipid homeostasis on neuronal function. Such findings would deepen the understanding of lipid regulation in neuronal physiology and its systemic implications.

5.        There are notable discrepancies in survival curves and experimental conditions described throughout the manuscript. Ensuring consistency in how data are presented and described (including exposure times and conditions in figures) will enhance the manuscript’s credibility and readability.

·     Line 191-192: It is not advisable to draw comparisons between transgenic worms expressing NHR-49 in cholinergic or glutamatergic neurons and those with pan-neuronal expression, as pan-neuronal expression was not included in this specific experiment. For greater accuracy, it should be noted that restoring NHR-49 specifically in cholinergic or glutamatergic neurons successfully rescues pathogenic lawn avoidance, contrasting with the results observed in nhr-49 mutants.

·     Figures 1, 2 & 3: There are noticeable inconsistencies in the survival curves of nhr-49 mutant nematodes across Figures 1, 2, and 3. It is crucial to address these discrepancies to ensure the reliability and reproducibility of the experimental results presented.

Minor Comments:

1.        Various sections of the manuscript display textual inconsistencies and formatting errors, particularly in the italicization of scientific names and gene names, which do not consistently adhere to C. elegansnomenclature standards. To ensure the manuscript meets the rigorous publication standards, meticulous proofreading and corrections are essential. Please ensure that all scientific and gene names are formatted correctly throughout the document (e.g., text, figures, graphs, figure legends etc).

·      The manuscript requires rephrasing for clarity and coherence at lines 32-33, 146-148, the sentence following 148, 154, 252, 255-256, 287, 341, and 347-348. Additionally, the legend for Figure 5 needs revision to enhance its clarity and descriptive quality. These changes will help maintain consistency and improve the overall readability of the text.

·      Italics Usage: Proper formatting, including the use of italics, is crucial for scientific names and gene names to align with standard scientific nomenclature. It is essential to apply italics correctly in lines 180, 277, 299, and throughout Figure 3 (including plots and legend). Ensuring consistent formatting will improve comprehension of the text and maintain the professional presentation standards.

2.        Clarifications in figure legends, particularly regarding what each line represents and ensuring alignment with the described experimental conditions, will improve the figure's interpretability.

·     Line 214 & Figure 3: The manuscript text refers to a 4-hour exposure to PA14, yet the legend for Figure 3 indicates a 10-hour exposure period. Clarification and consistency between the text and figure legends are essential to avoid confusion regarding the experimental conditions.

·     Figure 3C: To facilitate a more accurate comparison and enhance the interpretability of the results, it would be advantageous to include lawn occupancy data from the nhr-49 mutant in this figure.

·     Figure 3E: Similarly, incorporating lawn occupancy data for the strain expressing NHR-49 pan-neuronally in this figure would provide a clearer understanding of the effects and enhance comparative analysis across different experimental setups.

·     Figure 4B: The meaning of each line should be stated; it's important not to assume what each line represents based on previous figures.

·     The number of repeats each experiment was conducted has to be incorporated in all graphs or in their respective figure legends. The authors have included that only in some graphs. 

Comments on the Quality of English Language

The abstract contains several syntactical errors that may impede understanding and detract from the overall professionalism of the manuscript. It is essential for the authors to thoroughly revise and rewrite the abstract to ensure clarity, coherence, and grammatical accuracy. This revision will enhance the initial impression and accessibility of the study for readers.

Various sections of the manuscript display textual inconsistencies and formatting errors, particularly in the italicization of scientific names and gene names, which do not consistently adhere to C. elegans nomenclature standards. To ensure the manuscript meets the rigorous publication standards, meticulous proofreading and corrections are essential. Please ensure that all scientific and gene names are formatted correctly throughout the document (e.g., text, figures, graphs, figure legends etc).

·      The manuscript requires rephrasing for clarity and coherence at lines 32-33, 146-148, the sentence following 148, 154, 252, 255-256, 287, 341, and 347-348. Additionally, the legend for Figure 5 needs revision to enhance its clarity and descriptive quality. These changes will help maintain consistency and improve the overall readability of the text.

·      Italics Usage: Proper formatting, including the use of italics, is crucial for scientific names and gene names to align with standard scientific nomenclature. It is essential to apply italics correctly in lines 180, 277, 299, and throughout Figure 3 (including plots and legend). Ensuring consistent formatting will improve comprehension of the text and maintain the professional presentation standards.

Round 2

Reviewer 3 Report

Comments and Suggestions for Authors

The authors adequately addressed most of my concerns!